



# Assessing root-soil interactions in wetland plants: root exudation and radial oxygen loss

Katherine A. Haviland[1], Genevieve Noyce[1]

[1]Smithsonian Environmental Research Center, Edgewater, MD, 21401, USA

5    *Correspondence to*: Katherine Haviland (havilandk@si.edu)

**Abstract.** Plant rhizosphere processes, such as root exudation and root oxygen loss (ROL), could have significant impacts on the dynamics and magnitude of wetland methane fluxes, but are rarely measured directly. Here, we measure root exudation and ROL from *Schoenoplectus americanus* and *Spartina patens*, two plants that have had opposite relationships between biomass and methane flux in field experiments. We found contrasting rates of ROL

10   in the two species, with *S. americanus* releasing orders of magnitude more oxygen ($O_2$) to the soil than *S. patens*. At the same time, *S. patens* exudes high amounts of carbon to the soil, with much of that carbon pool reduced compared to exudates from other wetland species. This work suggests that the relative inputs of $O_2$ and carbon to the rhizosphere vary significantly between wetland plant species, potentially with major consequences on methane emissions, and highlights the importance of understanding how plant rhizosphere processes mediate soil

15   biogeochemistry at a community level. As global change drivers continue to impact wetlands, future research should consider how feedbacks from plant rhizosphere processes may exacerbate or mitigate coastal wetland methane emissions.



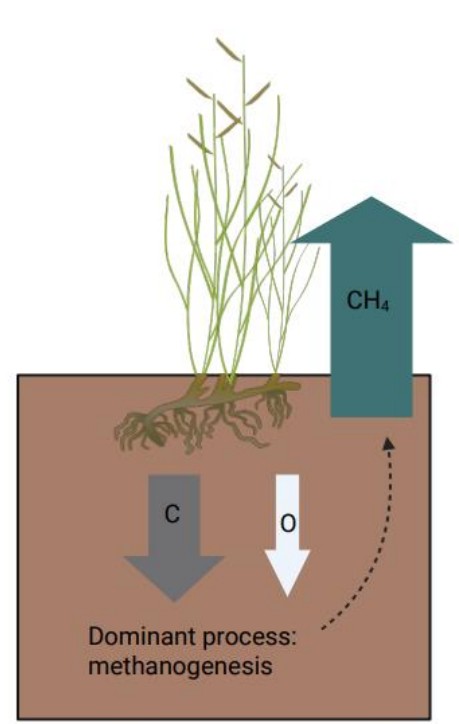
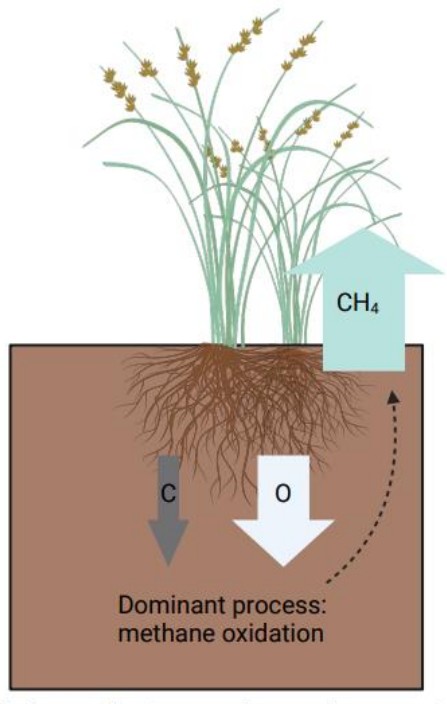

**Spartina patens-** greater root exudation, lower root oxygen loss: reducing effect on soil

**Schoenoplectus americanus-** lower root exudation, greater root oxygen loss: oxidizing effect on soil

**Graphical abstract**

## 1 Introduction

Rhizosphere processes such as root exudation and root oxygen loss (ROL, a.k.a. radial oxygen loss) can have major impacts on soil biogeochemical cycles and microbial activity (Hisinger et al., 2006; Haney et al., 2015). Root exudates include soluble organic compounds of varying molecular weights (Oburger & Jones, 2018), which may be readily used by microbes carrying out metabolic pathways including methanogenesis (Megonigal et al., 1999), nitrogen fixation (Li et al., 2021; Hanson, 1977), and sulfate reduction (Hines et al., 1999). Root exudates can have major influence on carbon cycling in vegetated soil, as up to half of plant-derived C is transferred to the rhizosphere (Kumar et al., 2006). Typically, root exudates have a net-reducing effect on soil redox conditions by fueling $O_2$ sinks directly through aerobic respiration, as well as driving sulfate reduction, an indirect $O_2$ sink due to the oxidation of the hydrogen sulfide end-product (Jensen et al., 2005; Martin et al., 2019). The exact molecular composition of root exudates is rarely studied (Oburger & Jones, 2018), but composition, in addition to total C input, is an important driver in how exudation may impact soil redox conditions. Additionally, the composition of the exudate pool can change depending on environmental contexts such as $CO_2$ concentration (Xiong et al., 2019) and temperature (Wang et al., 2021).



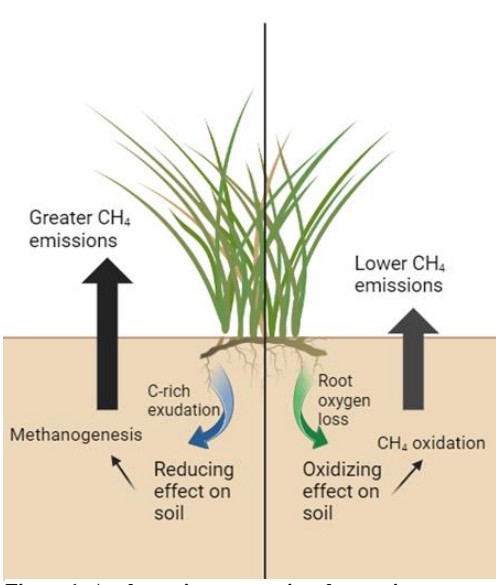

**Figure 1: A schematic representing the varying rhizosphere processes discussed in this paper and how they directly impact soil methane fluxes. Created in BioRender using graphics from the UMCES IAN media library.**

ROL also has a major impact on soil redox conditions, creating a more oxidized soil environment (Hemminga et al., 1988; Li et al., 2024), but the extent of this oxygenation varies at the species level, and changes in plant community composition can significantly alter sediment biogeochemical regimes (Koop-Jakobsen et al., 2021). ROL is an especially important process among wetland plants, which typically have very high rates of $O_2$ movement to the rhizosphere as an adaptation for growth in anaerobic, saturated soil (Lai et al., 2012). Because of the anaerobic state of wetland soils, $O_2$ derived from ROL is a major driver of oxidation in wetland soils (Craft, 2001). ROL is the net result of several plant traits related to gas exchange, such as aerenchyma tissue, and traits linked to ROL vary across and within species (Li et al., 2024). ROL is typically greater in daytime where photosynthesis and stomatal conductance are high (Lai et al., 2012). Many species of monocots develop strong barriers preventing $O_2$ loss under waterlogged conditions (Visser et al., 2000), though this barrier to ROL may come at the cost of nutrient uptake efficiency (Armstrong & Armstrong, 2005; Cheng et al., 2020). Currently, our understanding on the balance of root oxygenation to root exudation in wetland plants, and thus the resulting redox impact of plant roots on anaerobic soils (Fig. 1), is limited, despite the cascading effects on ecosystem biogeochemistry.

Wetlands can be a major sink or source of carbon (C) depending on environmental conditions, which has wide implications for global climate change (Noyce et al., 2023; Noyce & Megonigal, 2021; Mitsch et al., 2013; Lolu et al., 2020). Due in part to wetland plants' capacity for transporting carbon dioxide ($CO_2$) from the atmosphere to the anaerobic soils in which their roots are located (Kumar et al., 2006), wetlands may store soil organic C at very high rates, with some estimates putting them among the top C-storing terrestrial ecosystems (Valach et al., 2021; Nichols & Peteet, 2019; McLeod et al., 2011). However, many wetlands are a source of C to the atmosphere through enhanced production of methane ($CH_4$), the rates of which may depend on soil environmental conditions, which are influenced by root exudation (Noyce & Megonigal, 2021). As global change impacts the productivity and community composition of wetland plants, rhizosphere processes will likely change as well (Jiang et al., 2020; Zhai et al., 2013), exacerbating species-level differences in root exudation and oxygenation.

As sea levels rise, *Schoenoplectus americanus*—a salt- and flood-tolerant $C_3$ sedge—has encroached into areas on the Atlantic coast of the United States formerly dominated by *Spartina patens*, a $C_4$ grass (Mueller et al., 2020). Additionally, *Phragmites australis* (ssp. *australis*), an invasive reed, has encroached on native wetland plant habitat in the region (Sciance et al., 2016). We have previously proposed that species-level differences in rhizosphere



processes drive observed differences in wetland soil biogeochemistry, where *S. patens*-dominated areas have higher methane emissions than *S. americanus*-dominated sites (Noyce & Megonigal, 2021; Mueller et al., 2020), but direct measurements of root exudates and ROL remain elusive in a field environment. Here, we conduct a growth chamber study to identify root-associated metabolites and root exudation rates from these three species (as well as *Spartina*

*alterniflora,* another common wetland plant), and estimate ROL in *S. patens* and *S. americanus.* For further experimentation on ROL and the effect of exudates, we focus on *S. patens* and *S. americanus*, the two most dominant plants at Kirkpatrick Marsh. We follow with a corresponding incubation experiment to estimate the role that exudate identity plays in methane production in wetland soils colonized by either *S. patens* or *S. americanus.* Based on past field data showing greater methanogenesis in *S. patens*-dominated areas (Mueller et al., 2020), we

hypothesized *S. patens* will have lower rates of root oxygenation, and higher rates of root exudation, than *S. americanus,* and that the cocktail of exudates coming from the roots of *S. patens* will drive higher rates of methanogenesis compared to the *S. americanus* exudate pool.

## 2 Methods

### 2.1 Growth conditions

We harvested *Schoenoplectus americanus* and S*partina patens* rhizomes by hand from Kirkpatrick Marsh, located on the Rhode River, a sub-estuary of the Chesapeake Bay, MD, USA, in early July 2023. We collected *Phragmites australis* (subspecies *australis*) rhizomes in the same area in August 2023 and acquired *Spartina alterniflora* plants from Delmarva Native Plants (Hurlock, MD). We placed rhizomes in shallow trays containing Dakota reed and

sedge peat hydrated with DI water. Once plants from these rhizomes reached >20 cm tall, we carefully separated individuals from each other with at least 3 cm rhizome and moved them into rhizoboxes (20 x 3 x 30 cm (WxDxH), inside dimensions) purchased from Vienna Scientific (Alland, Austria) containing the same peat. We monitored growth in the rhizobox for 1 week by measuring plant height every 2-3 days (Fig. S1) before all analyses, keeping soil consistently moist. Rhizoboxes were half-submerged in distilled water adjusted to near site salinity (~10 ppt)

with InstantOcean between samplings. Ten plants from each species at a time were acclimated in the rhizoboxes for 1 week in growth chambers at the Smithsonian Environmental Research Center. Full-spectrum growth lights (~400 µmol m$^{-2}$ s$^{-1}$ PPFD) in the growth chamber were on during daytime for 16 hours, and off during nighttime period for 8 hours during the root exudate phase of analysis (August-December 2023), and switched to 12-12 h during root O$_2$ analysis (January-April 2024). Temperature was set to 25 ° C during the light period (referred to as "day"), and 18 °

C during the dark period (referred to as "night"). Humidity was not manipulated, but was typically higher in the day period due to temperature (70-85%), and lower (60-70%) during the night period. We tracked humidity and temperature using three separate devices: a HOBO Onset logger situated above the plants; a custom-built Arduino microcontroller connected to K30 (Senseair) and BME680 (Adafruit) sensors (measured ambient CO$_2,$ temperature, air pressure, humidity, and volatile organic compounds); and a wall-mounted Titan Saturn 5 Digital Environmental

Controller.





**Table 1: Species used in this study. A greater number of analyses were carried out on *S. americanus* and *S. patens* because they are the most abundant species at Kirkpatrick Marsh, and have been targets of prior research on greenhouse gas dynamics (Mueller et al., 2020; Noyce & Megonigal, 2021).**

| Species | Photo-synthetic pathway | Notes | Analyses |
|---|---|---|---|
| *Schoenoplectus americanus* | $C_3$ | Salt-tolerant sedge expanding in many formerly *S. patens* dominated regions (Mueller et al., 2020) | Exudation, ROL, and incubation |
| *Phragmites australis* | $C_3$ | Invasive reed expanding in many areas of the US Atlantic coast (Sciance et al., 2016) | Exudation only |
| *Spartina patens* | $C_4$ | High marsh grass with limited salt tolerance (Mueller et al. 2020) | Exudation, ROL, and incubation |
| *Spartina alterniflora* | $C_4$ | Well-studied and salt-tolerant low marsh grass (Hester et al., 2001) | Exudation only |




## 2.2 Root exudation

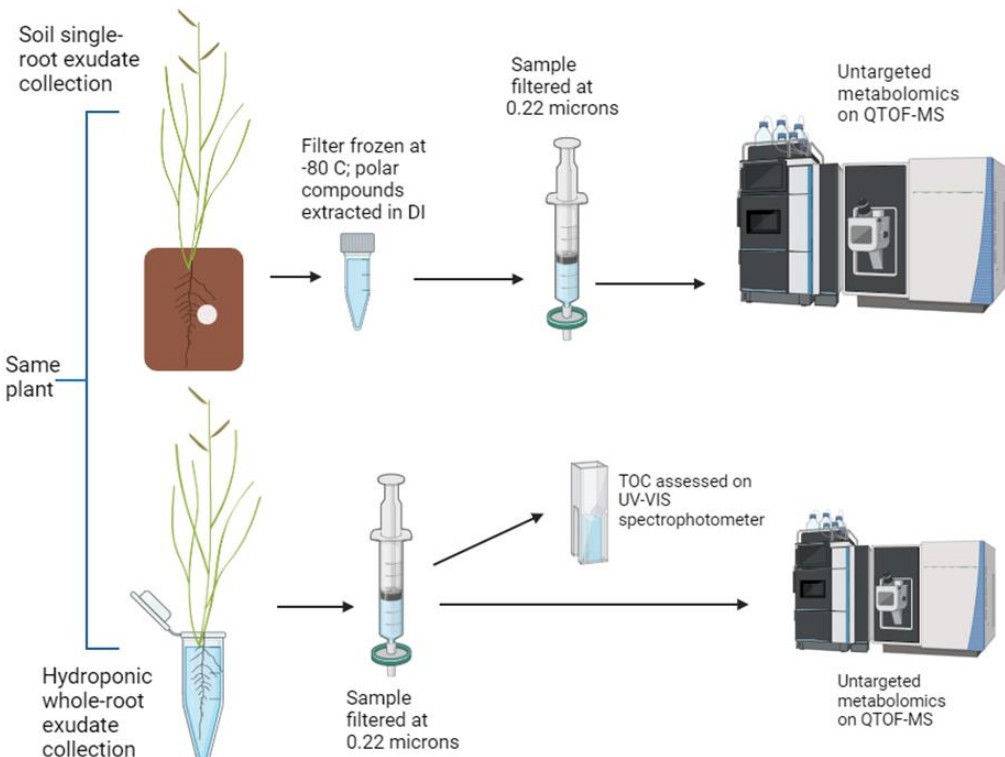

**Fig. 2: Root-associated metabolite collection and analysis schematic for one plant. A total of forty samples were processed. Figure created using BioRender.**

Root-associated metabolites were collected using a combination of soil and hydroponic methods (Fig. 2). All methods were non-sterile for all or some of the sampling period, meaning some identified metabolites are related to soil and rhizosphere microbial processes. We will use the phrase 'root-associated metabolites' to refer to all products detected using both methods, which likely include root exudates and rhizosphere soil microbial products. For the soil method, an individual adventitious root, between 5-15 cm in length, was placed between two pieces of clean

unbleached wax paper within the rhizobox, while the rest of the roots were left in the soil. We noted the root's color, depth in the soil, and presence or absence of hairs. The chosen root was thoroughly rinsed with DI to remove soil, and gently patted dry with a Kimwipe. We then placed a Whatman GFF (25 mm diameter) filter saturated with DI water over top the root (extended from the root tip through apical zone) between the two pieces of wax paper (Neumann, 2007). The rhizobox was sealed and returned to the growth chamber under the growth lights for ~1.5 h

before the filter was retrieved and immediately frozen at -80 °C prior to analysis. We repeated this process three times during the day (beginning >2 h after lights turned on), and three times during the night period (beginning >5 h prior to lights turning on), for each of ten replicate plants per species. On the day of sampling, each plant's height,



root depth, sampled root depth (i.e. the depth at which the filter was placed), and the color of the sampling root was observed. Following sampling, plants were harvested and dried for aboveground and belowground biomass.

Following filter analysis, plants were removed from the soil-filled rhizobox, and the roots were thoroughly (but carefully, to prevent tissue damage) washed with deionized (DI) water. Roots were then allowed to soak in DI water overnight, before being rinsed again. We then sampled root-associated metabolites hydroponically in 30 mL DI water in sterile containers for 1.5 h. Hydroponic root-associated metabolites were filtered (0.45 micron) and separated into two portions, with one aliquot for metabolomic analysis, and one aliquot to be assessed for total

organic carbon (TOC) using a UV-VIS spectrophotometer at 254 nm after Oburger et al. (2022). A subset of data collected via TOC spectrophotometry was also assessed on a Shimadzu TOC analyzer (Fig. S2), and spectrophotometrically derived TOC values were adjusted according to that relationship.

For metabolomic analysis, polar root-associated metabolites were extracted from filters by soaking in ultra-pure LCMS grade distilled water, followed by centrifuging at 7900 rpm for 15 min at 4 °C. The supernatant was then

passed through a 0.22-micron filter, and again frozen at -80 °C. All metabolomics samples were brought to the University of Maryland Baltimore County (UMBC)'s Molecular Characterization and Analysis Complex (MCAC), where they were processed for untargeted metabolomics on a quadruple time-of-flight (QTOF) mass spectrometer (MS) equipped with a Dionex UltiMate 3000 UHPLC. For each sample, two aliquots were assessed, first in negative and then in positive ionization mode. Compound identification was carried out by MCAC using Bruker

MetaboScape software, based on m/z peak, retention time, and score, with the Bruker MetaboBASE Plant Library, and the Natural Products Atlas (NPA). Additional details on the untargeted metabolomics workflow is reported in the supplementary materials. Compounds were selected for further analysis as potential root exudates if they had been reported in Reference Metabolome Database for Plants (RMDP, https://www.biosino.org/RefMetaDB/; Shi et al., 2024) as a known plant metabolite. For all compounds, we calculated nominal oxidation state of carbon (NOSC)

after Gunina & Kuzyakov (2022).

## 2.3 Incubation

Incubations were conducted on soil collected from 2-10 cm depth within a *P. australis*-dominated area of Kirkpatrick Marsh in August 2023. We removed all large roots and debris from the soil, and then homogenized it. We collected a root exudate 'stock' solution from each of two target plant species: *S. patens*, and *S. americanus*,

using 10-20 plants from each species. Roots from the plants were thoroughly cleaned and placed in 100 mL DI water as in the hydroponic sampling above. TOC in the stock solution was tested on the spectrophotometer as above, and diluted with DI to reach a final concentration of 4 mg l$^{-1}$ TOC of the stock solution for both species. We combined 3 g wet soil with 10 mL DI water in 15 serum vials, and flushed the vials with $N_2$ for 15 min to ensure the headspace was anaerobic. We placed vials in a dark incubation chamber set to 25 °C for 2 weeks to acclimate.

Samples were kept on a shaker at 90 rpm to keep sediment from settling out. At the end of 2 weeks, we measured baseline $O_2$ to ensure an anoxic environment within the jars, and initial headspace methane. We measured $O_2$ using a PyroScience Firesting sensor and 500 µm needle-tip $O_2$ minisensor. We measured methane by removing 3 mL headspace from each jar and injecting it into a Shimadzu Nexis GC-2030. We then randomly assigned each serum



vial to one of three groups: *S. patens*, *S. americanus*, or control, taking care to ensure that headspace baseline average methane and $O_2$ was consistent among the three groups. In week 1, we added 1 mL stock concentration of their given species (or an injection of DI in the control group) daily for 5 d, followed by methane and $O_2$ analysis on the fifth day as above. In week 2, samples received a daily addition of 1.5 mL of stock, followed by analysis; and in week 3, samples received a daily 2 mL addition of stock. We then waited 2 weeks with no stock additions, and carried out a final gas sampling. At the end of the experiment we sampled the slurry for dry mass and loss on ignition (LOI) at 450 °C for 4 hours.

**2.4 Oxygen**

We assessed location of root oxygenation within the root mass of *S. patens* and *S. americanus* using the methylene blue method after Armstrong & Armstrong (1988) in 3 plants of each species. We used the results of this method to determine the extent of barriers to ROL in each species and inform areas of the rooting mass to focus on for planar optode analysis, described below.

We used 2-D planar optode imaging to characterize and quantify ROL in *S. americanus* and *S. patens*. We imaged root oxygenation using a PreSens Precision Sensing (Regensburg, Germany) VisiSens TD camera setup in the same growth chamber environment. We carried out weekly two-point calibrations with anoxic soil (0%) and water in equilibrium with atmosphere (100% atm $O_2$). We measured root $O_2$ in rhizoboxes containing individual samples of either *S. americanus* or *S. patens* (n = 8 for each species) in Dakota reed and sedge soil. For some additional *S. patens* plants (n = 5), we carried out further imaging on roots grown in rhizoboxes on soil that was autoclaved at 121°C with a triple pre-vacuum cycle in an attempt to reduce microbial sediment $O_2$ demand and better quantify ROL. For a full rationalization of the use of planar optode technology to image wetland plant radial $O_2$ loss, see Koop-Jakobsen et al. (2018), and Jiménez et al. (2021). Each plant was grown in 12 h light 12 h darkness, and the overhead lights went off for 1 minute as images were collected every h for 3-4 days per plant. The first 24 h period after the $O_2$ -sensing optode foil was placed was excluded from the dataset. Per plant, this resulted in at least 24 images captured in light, and 24 in darkness (72 h cycle, subtracting the first 24 h). Notably, using this method, $O_2$ can only be imaged if the rate of $O_2$ release exceeds the rate of sediment $O_2$ demand (SOD). Planar optode imaging of root $O_2$ in a rhizobox environment has been shown to overestimate ROL somewhat (Frederiksen & Glud, 2006), but is still useful as an estimate, especially for inter-species comparison of data collected using the same method.

Images were exported from the VisiSens software and initially analyzed for change in R using the raster analysis package 'terra' (Hijmans, 2023). For each plant, we assessed change in average pixel value from the day compared to the night period, and standard deviation of pixels across each 24 period. We measured the average % $O_2$ in both day and night periods around the root cap for each plant, and recorded the peak and average $O_2$ concentration in root-influenced regions (regions of interest, or ROIs) in the light and dark periods, as well as average $O_2$ concentration in comparable ROIs in the background bulk soil. Additionally, we periodically measured soil $O_2$ demand (SOD) in the test soils by injecting known quantities of distilled water in equilibrium with atmosphere (~21% $O_2$) at the soil-foil interface and measuring time for complete consumption of the $O_2$.



Before and after foil placement, we took RGB images of the rooting zone, and created root diagrams by tracing root
area behind the foil. From these diagrams, we calculated root surface area assuming root diameter based on root
width in the images (treating roots as cylinders). We combined SOD measurements with differences in % $O_2$
between background conditions and root conditions to estimate the rate of $O_2$ release from the plant root in the soil.

**2.5 Statistical analysis and data availability**

Differences in the root exudate profiles produced by the four different plant species were assessed using principal
components analysis (PCA) carried out in R, and top contributing root-associated metabolites to differences between
groups (species, light vs. darkness, sampling method) were identified using the package 'factoextra' (Kassambara &
Mundt, 2020). Single-factor species-level differences (TOC, average NOSC) were assessed by Tukey's HSD or
Dunn test (Dunn test carried out using package 'rstatix' [Kassambara, 2023]), depending on the distribution of data,
assessed by Shapiro-Wilke test. Initial analyses (network analysis, heatmap creation) and data exploration were
carried out using MetaboAnalyst 6.0 (https://www.metaboanalyst.ca), though all final data analyses were carried out
in R. Raw mzML files of all detected root-associated metabolites in the 4 plants is freely available and hosted in the
MassIVE database managed by UC San Diego's Center for Computational Mass Spectrometry (MassIVE ID
MSV000094652, doi:10.25345/C55D8NR99). Incubation data (headspace methane) over time was assessed via
linear model with treatment (*S. americanus, S. patens*, or control) as a block factor, and with Tukey's t-test
comparing the three treatments at each time point. Differences in ROL per plant species was assessed by t-test, after
checking for normality.

**3 Results**

**3.1 Root exudation**

Metabolites identified using both methods included amino acids, organic acids, sugars, fatty acids, and secondary
metabolites. We saw greater species-level separation in the hydroponic metabolites across all compounds (Fig. 3),
but greater species-level separation in the single-root soil filter metabolites when considering only compounds
previously reported as plant metabolites. Compounds that differed most between species in the all-metabolites
dataset included secondary metabolites with antifungal and antibacterial properties. Compounds that differed most
by methodology included aspartic acid, which was found in the hydroponic samples for all species, but not the filter
samples. For variation at the species level, sucrose was found more commonly in samples collected on filters, and
was at the highest levels in *P. australis*, and identified in some individuals in both *Spartina* species, but rarely
detected in the *S. americanus* root exudate pool. Among root-associated metabolites commonly reported as root
exudates, those that drove species-level variation in the hydroponic dataset included aspartic acid (8% contribution
to PCA-2) and sucrose (7.3%), and aristeromycin M (11 % contribution to PCA-1) and histidine (4%). Aspartic acid
was highest in *S. americanus*, aristeromycin M was greatest in *P. australis*, and thymine was greatest in *S.
alterniflora*. Other compounds with significant differences by species (identified using MetaboAnalyst) are shown in
Figs. S5-6. Of the 642 compounds identified, we flagged 75 previously reported as plant metabolites in RMDP. The



75 known plant metabolites detected included those commonly seen within root exudate pools such as sucrose, succinic acid, pyruvic acid, valine, alanine, leucine, phenylalanine, arginine, and aspartic acid (Vranova et al., 2013) as well as nicotinic acid, salicylic acid, and benzoic acid (see Table S1 for full list). All compounds discovered in the root-associated metabolite pool fell into the following classes (ordered by relative abundance): organoheterocyclic compounds; lipids; nucleosides, nucleotides, and analogues; organic oxygen compounds; isobenzofurans; benzenoids; organic acids; organic nitrogen compounds; phenylpropanoids and polyketides; coumarins; and phenols. When classed by compound type, there were significant differences at the species-level in relative abundance for benzenoids (*S. patens* > *S. alterniflora* and *P. australis*), coumarins (*S. patens* > *P. australis*), and organic acids (*S. alterniflora* > *P. australis*; *S. patens* > all).

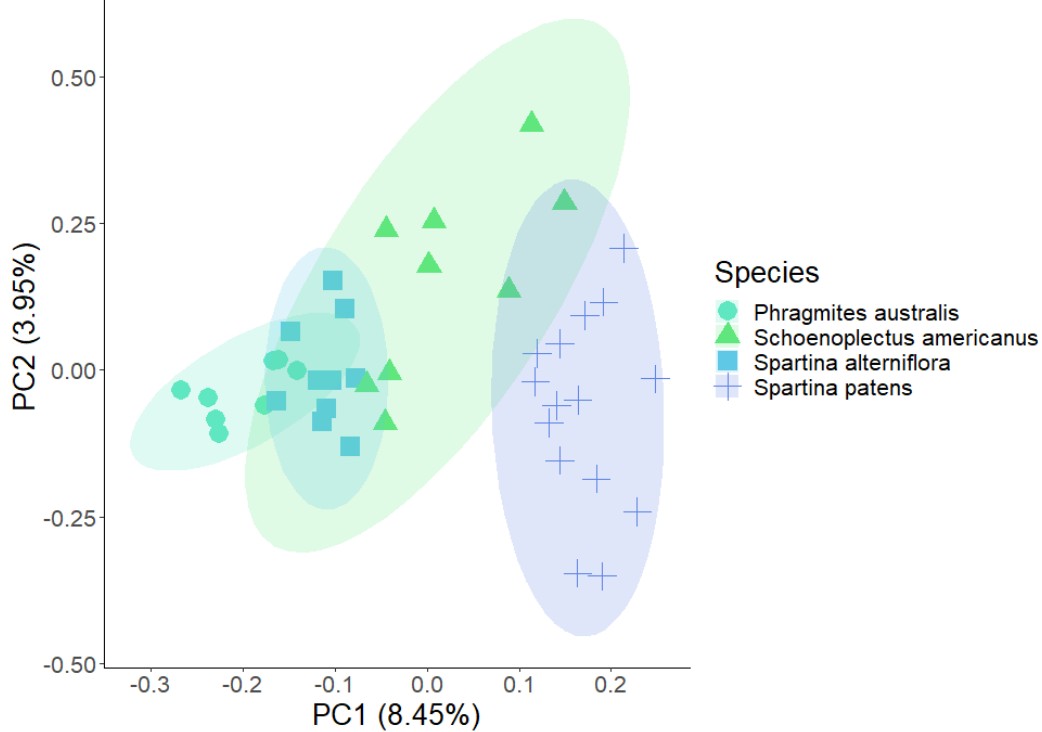

**Fig 3. PCA on all root-associated metabolites ($n$ = 647) collected hydroponically from wetland plant roots during daytime. Each point represents one replicate of a species.**

Total TOC exudation rate per root biomass was greater in *S. patens* than any other species (p < 0.05, Dunn's test). *P. australis* exudation rate was lower on average, but showed greater variability, while *S. alterniflora* and *S. americanus* had similarly low exudation rates (Fig 4). The average NOSC of the exudate pool(weighted by amount of each compound exuded) was lowest in *S. patens* (p < 0.05, Tukey's HSD test), and highly constrained across the other three species (Fig 5).



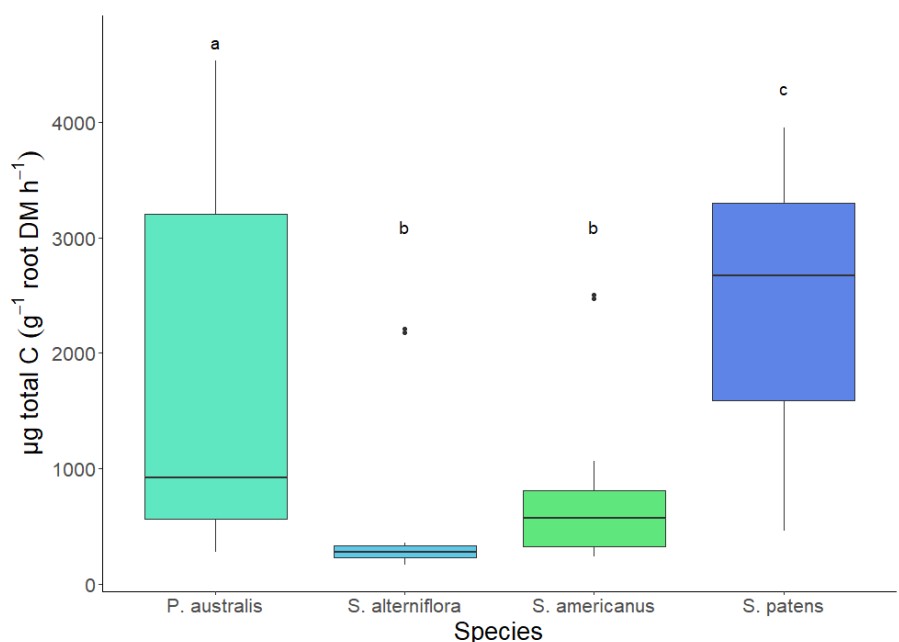

**Fig 4. Hydroponic TOC exudation rate of 4 wetland species (*Spartina patens, Phragmites australis, Schoenoplectus americanus,* and *Spartina alterniflora*) in daytime. Letter denotes significance ($p < 0.05$) via Dunn test (non-parametric).**

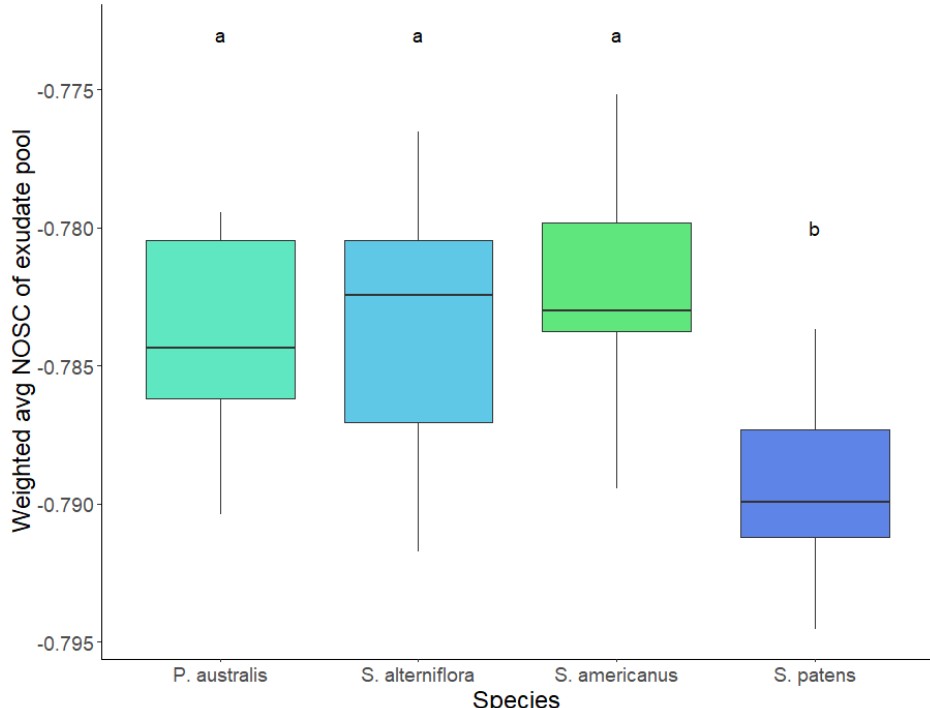




**Fig 5. Average NOSC of hydroponically sampled root exudates in daytime, weighted by relative abundance of compound exuded. Letters denote significance ($p < 0.05$) by Tukey's HSD test.**

Diel patterns of root-associated metabolites collected using the soil filter method depended on species. In *Spartina alterniflora*, relative abundance (calculated from peak intensity) of all detected exudates summed was greater in the day ($p = 0.03$), while in all other species, there was no difference in relative abundance of detected exudates based on time ($p = 0.8$ for *S. patens*; $p = 0.9$ for *S. americanus*; $p = 0.09$ for *P. australis*). In all plants, there was lower diversity of root-exudate compounds during the night (Fig S3).

**3.2 Incubation**

Daily addition of root-associated metabolites of any species to the peat slurry resulted in increased headspace methane compared to the control groups. The response varied the greatest in *S. americanus*, which had total headspace methane values ranging from 0 – 16,000 ppm across the experiment (Fig. 6). In weeks 1 and 2, there were no significant differences between either treatment and the control. In weeks 3 and 5, headspace methane in *S.*
*patens*-exudate treated jars was significantly greater than the control ($p < 0.05$), and *S. americanus*-exudated treated jars were somewhat greater than the control ($p < 0.1$). There was never any difference between the *S. americanus* and *S. patens* treatments.

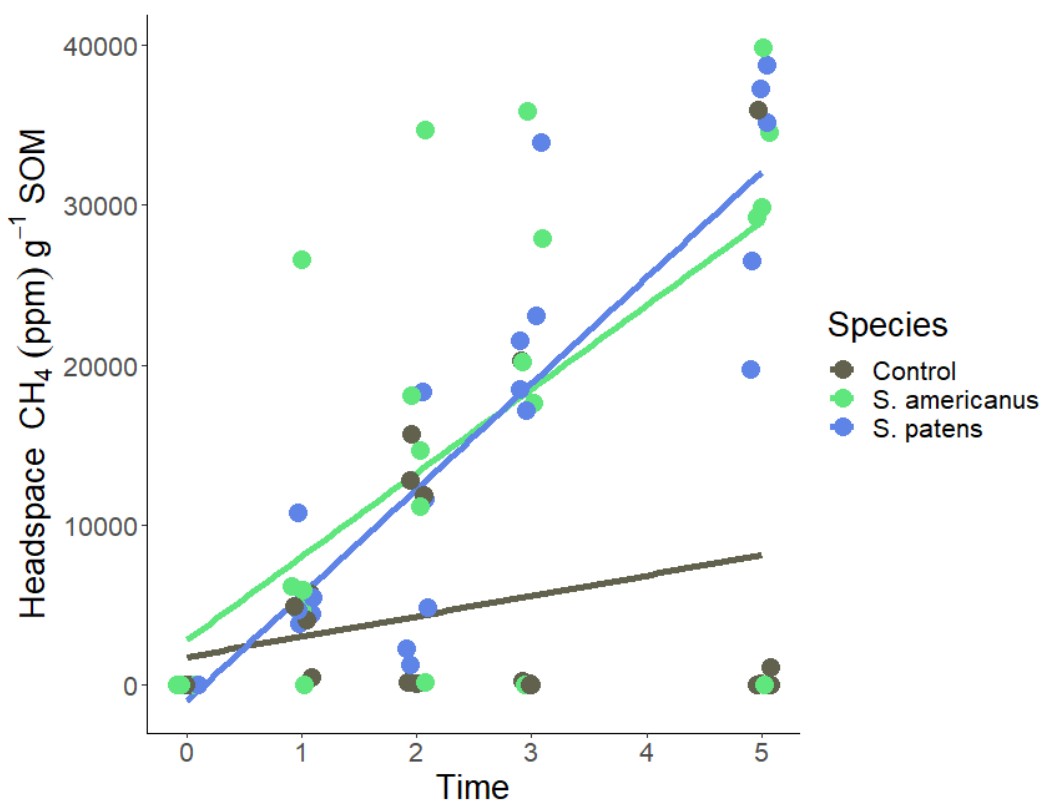





**Fig. 6: Headspace methane (ppm) normalized to soil organic matter (SOM) over time (weeks) in incubation serum vials. Vials contained site soil and DI in slurry. 1 mL of root-associated metabolite stock solution collected hydroponically from plants (or DI for controls) was added to each vial per day for 5 days in week 0-1, 1.5 mL per day in week 1-2, and 2 mL per day in week 2-3. Nothing was added to the vials in weeks 3-5.**

**3.3 Oxygen**

Methylene blue analysis revealed greatest $O_2$ release from the tips of adventitious roots in *S. americanus,* and from the fine roots and root hairs in *S. patens* (see Fig. S8 for images showing typical patterns of oxygenation). In both species, young light-colored roots consistently showed the greatest oxygenation. Most *S. patens* roots showed little barrier to ROL compared to *S. americanus*, which had regions of blue found only at root tips.

When assessed during daytime, soil $O_2$ demand (SOD) in the rhizobox bulk soil (Dakota reed and sedge peat) was high. Accounting for soil porosity of 0.45, SOD occurred at a rate of 812 nmoles $L^{-1}$ $s^{-1}$. At night, at lower temperatures, the rate was 108 nmoles $L^{-1}$ $s^{-1}$. SOD was higher deeper in the soil column. Based on this, we focused our planar optode imaging efforts on the top 10 cm of soil. In the autoclaved peat, there was less of a difference between SOD in the night and day periods, and SOD was much lower overall compared to the un-sterilized peat

(Table 2). However, once an *S. patens* plant was added to the rhizobox, SOD increased rapidly over the acclimation period prior to ROL analysis, and was on par with the un-sterilized peat within 3-5 days.

**Table 2. Average SOD in the rhizoboxes. The SOD of sterilized peat is comparable to un-sterilized levels within 3 days of planting.**

| Soil | SOD, day (nmoles $L^{-1}$ $s^{-1}$) | SOD, night (nmoles $L^{-1}$ $s^{-1}$) |
|---|---|---|
| Peat only | 812 | 108 |
| Sterilized peat before planting | 68 | 41 |
| Sterilized peat with *S. patens* plant | 540 | 180 |


In *S. americanus*, zones of ROL were most evident on young adventitious roots that were white in color, as in the methylene blue data. Not all roots within the extent of the optode foil had associated ROL. Of the 16 plants imaged over 3 24-h light-dark cycles, we collected 9 regions of ROL (7 roots across 5 plants from *S. americanus*, 1 plant with 2 roots in *S. patens*) that qualified for further analysis (associated with a root, smooth soil surface without

macropore present, diel pattern repeating at least once).

In the majority of *S. americanus* roots analyzed (6 of 7), we saw slightly greater buildup of $O_2$ in root-associated regions at night. $O_2$ in regions around roots associated with oxygenated zones built up to highest levels near dawn, and dropped to lowest levels around dusk, but was always greater than background levels. This probably reflects greater sediment $O_2$ demand in the daytime driven by increased daytime temperatures in the growth chamber.

Background (bulk soil) $O_2$ followed a similar diel pattern, though less pronounced than around the roots. However,



there is likely much greater microbial biomass around roots due to the inputs of carbon and $O_2$ fueling microbial processes, so $O_2$ consumption in the rhizosphere region may be higher than in the bulk soil. In *S. patens,* we rarely observed regions of $O_2$ build-up behind the optode foils, implying that for most roots, although ROL occurs (as demonstrated by the methylene blue results), it never exceeded the rate of SOD (Table 2, Table 3) in the rhizoboxes.

However, for 1 plant of *S. patens* (of 8 assessed on un-sterilized soil), with the optode foil placed on two thicker-than-average roots, $O_2$ buildup was detectable at similar levels of *S. americanus* in peat. Accounting for root area (estimated from root length and diameter in images with area calculated assuming roots as cylinders), ROL in *S. americanus* ranged from $65 – 621$ nmol m$^{-2}$ s$^{-1}$ and was much greater on average during the high temperature day period (average = 391 nmol m$^{-2}$ s$^{-1}$) than the night period (average = 62 nmol m$^{-2}$ s$^{-1}$). *S. patens* ROL was largely not

detectable in the high SOD peat background, implying ROL below the rate of SOD, but for the one plant that had detectable ROL in peat on two roots, the rate was much greater during the daytime, at 166 nmol m$^{-2}$ s$^{-1}$, and remained undetectable at night. Although autoclaving peat did initially reduce SOD, *S. patens* ROL was similarly undetectable on this growth medium during both day and night on all but one plant, due to the increase in SOD after introducing a plant (Table 2). Note that, because not all roots showed signs of ROL during methylene blue testing,

our numbers when scaled up to the per root area level likely represent an overestimation.

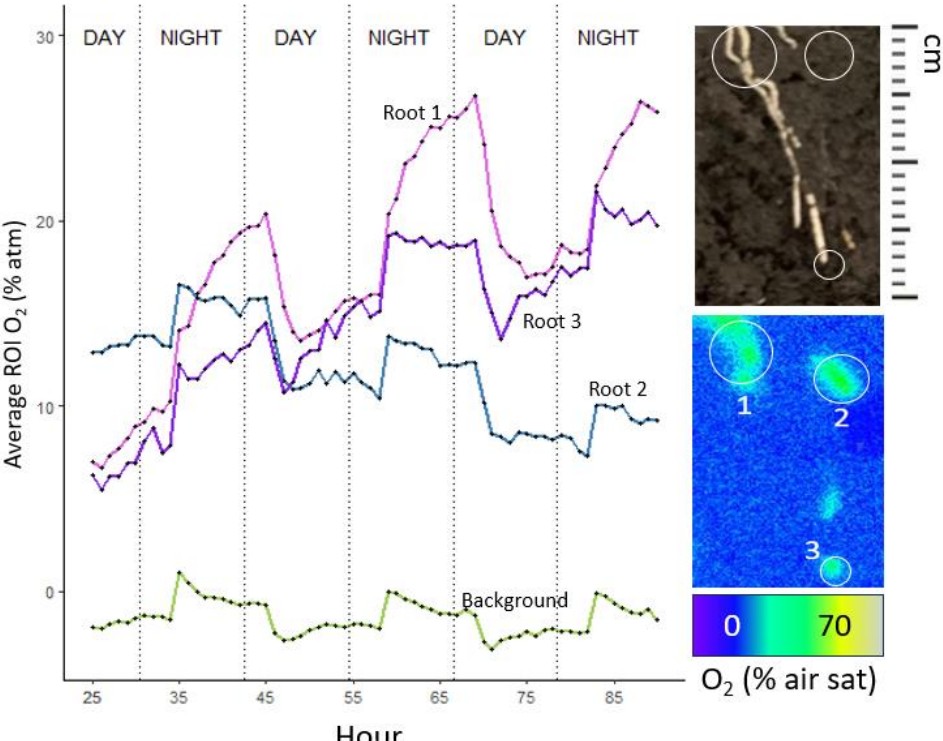

**Fig. 7: Data from a characteristic *S. americanus* plant with several roots showing ROL over 72 h, with images taken every hour. Panel images at R show region of interest around the root during select time periods, as well as an image of the**






roots located in the ROI. Note that root images were taken prior to the 72-h collection, and exact locations of roots likely
moved as growth occurred over the 4-d period.

**Table 3: Average (and standard deviation) O₂ data from each species for *n* roots. ROL was determined based on the size of the oxygenated zone within a stable period of at least 3 h. Images from plants without oxygenated zones (*n* images = 3 for *S. americanus*, *n* = 11 for *S. patens*) were excluded. % O₂ in ROI refers to root-associated O₂ (at least 3 h) as % atm on peat soil, or in an 80-20 sand-peat mix (for *S. patens* only, in order to quantify ROL in lower SOD conditions).**
**(*) The *S. patens* plants likely represent maximum values, as all other plants had undetectable ROL.**


| Species | $N$ | O₂ (% atm) in ROI | | ROL (nmoles m⁻² s⁻¹) | |
|---|---|---|---|---|---|
| | | Day | Night | Day | Night |
| *Schoenoplectus americanus* | 7 | 19.2 (16) | 23 (19) | 357 (117) | 62 (26) |
| *Spartina patens* | 3* | 38.5 (19) | 1 (0) | < 154 (21) | < 31 (29) |

## 4 Discussion

### 4.1 Combined effects of carbon and oxygen

Root exudation and ROL varied across species, with *S. patens* exhibiting the greatest root exudation rate, as well as the most reduced rhizosphere-associated metabolite pool, while *S. americanus* consistently showed greatest oxygenation of the rhizosphere. Our data clearly demonstrate that *S. patens* exudes a greater amount of C per O compared to *S. americanus*. Interestingly, the composition of the rhizosphere-associated metabolite cocktail alone does not result in different methane yields between the two species (Fig. 5), but differences in the rate of exudation

overall, or over longer time scales, plus the addition of varying rates of soil oxygenation by roots, likely has differential effects on soil biogeochemistry between the two species. At a marsh-scale, these processes are altered by sea level rise resulting in community change from *S. patens* to *S. americanus* in mesohaline regions of Chesapeake Bay (Mueller et al., 2020), with increased ROL and reduced root exudation in *S. americanus* likely leading to lower methane fluxes from the soil and representing a negative feedback cycle.

We've previously shown that field-scale methane fluxes are higher from sites dominated by *S. patens* compared to sites dominated by *S. americanus* (Noyce & Megonigal, 2021). Similarly, prior work has also found higher net methane emissions from mesocosms with *S. patens* compared to mesocosms with *S. americanus* (Mueller et al., 2020) This has led to a hypothesis that root actions of *S. americanus* act as a 'net oxidizer' on the soil, reducing methane emissions through aerobic methane oxidation, while *S. patens* has the opposite effect as a 'net reducer'

(Noyce & Megonigal, 2021). Our results support the mechanisms underlying this theory, with *S. americanus* exuding the lowest TOC per unit biomass of any species studied, and *S. patens* the most (Fig. 3), while *S. patens* also released considerably less O₂ from roots than *S. americanus*. Some of the lower rates of ROL seen in *S. patens* could be a result of greater microbial respiration driven by the increased TOC release, though further research would be needed to evaluate how these two processes impact the microbial community at the rhizosphere level.

Nevertheless, the lack of persistence of O₂ in the *S. patens* rhizosphere shows that *S. patens* clearly has a less





oxidizing effect on the soil than *S. americanus*, where $O_2$ released from roots persists through the day and night period. Additionally, the NOSC of the rhizosphere-associated metabolite pool from *S. americanus* is more oxidized, which may result in a less reduced rhizosphere environment even in the absence of ROL, again lowering methanogenesis and thus methane flux from the soil. *S. patens* had the lowest NOSC of any the 4 species studied,

implying that the TOC pool coming from the roots of *S. patens* is more reduced overall, likely leading to more reducing soil conditions. This more reduced pool may also be a greater sink for $O_2$ released by roots, resulting in the inability to image ROL in the majority of the *S. patens* samples.

### 4.2 Comparison to other studies

Input rates of total organic carbon (ranging from 200-4300 µg C g$^{-1}$ root DM h$^{-1}$) varied between the 4 species on a

per root biomass area, but were comparable to other submerged plant species such as *Oryza sativa* (rice), which typically exudes between 500-4500 µg C g$^{-1}$ root DM h$^{-1}$ (Aulakh et al., 2001; Xiong et al., 2019). Detected compounds across species were also similar to those found as root exudates in other systems, with organic acids making up the majority of detected compounds that differed on a species level, similar to *Typha latifolia* and *Vetiver zizanioides* (Wu et al., 2012). Unlike some other studies of root exudates in other wetland species (Wu et al., 2012;

Chen et al., 2016), we did not detect oxalic acid in our rhizosphere-associated metabolite pool for any species, possibly due to methodological differences, as both the other studies were conducted on plants grown entirely under hydroponic conditions. Growth conditions have been shown to play a significant role in structuring the root exudate composition of various plants, and many of the compounds and compound classes we found in our set-up are similar to other metabolomic profiles of the root exudate pool collected using a nonsterile set-up (McLaughlin et al., 2023).

Organic acids commonly found in our rhizosphere pool were nicotinic acid (reported in *Arabidopsis thalania* root exudate pool by Pantigoso et al., 2020), succinic acid and fumaric acid (both reported in maize exudate pool by Nardi et al., 1997), and pyruvic acid (reported along with fumaric acid in agricultural grasses by Maurer et al., 2021). Like other studies (reviewed in Vranova et al., 2013), commonly found amino acids in our samples included aspartic acid (found at highest levels in *S. americanus*), and leucine, valine, arginine, and histidine, all of which

occurred equally across the four species. Addition of amino acids to soil has been shown to reduce methane yield (Liu et al., 2021), and overall, amino acid relative abundance was higher in *S. americanus* than other species (Table S1). Organic acids, which were highest in the *S. patens* rhizosphere (Fig. S4), have been shown to increase methane yield (Girkin et al., 2018). These variations in the composition of the root exudate cocktail may affect biogeochemical conditions in the rhizosphere, favoring methane production in *S. patens,* and decreasing methane

production in *S. americanus*, on longer time scales than we assessed in our incubation experiment.

ROL rates in *S. americanus* are comparable to results from other plants using this method, such as *Spartina anglica*, which Koop-Jakobsen & Wenzhöfer (2015) found to have a ROL rate of 249-300 nmol m$^{-2}$ s$^{-1}$, slightly lower than our measured rate in *S. americanus* (357 nmol m$^{-2}$ s$^{-1}$), and higher than our measured rate in *S. patens* (>154 nmol m$^{-2}$ s$^{-1}$). Colmer (2003) found similar rates of ROL to our values for *S. patens* at root apices of *O. sativa* (175 nmol m$^{-2}$

s$^{-1}$), and similar to *S. americanus* in *P. australis* (390 nmol m$^{-2}$ s$^{-1}$), *Hordeum marinum* (273 nmol m$^{-2}$ s$^{-1}$), and *Phalaris aquatica* (300 nmol m$^{-2}$ s$^{-1}$); all of these wetland species showed a 'tight' or 'partial' barrier to ROL,



similar to what we saw in *S. americanus* using methylene blue (Colmer, 2003). Conversely, *S. patens* showed less of a barrier to ROL on many roots in the methylene blue images, and had lower ROL in optode images as well. In most *S. patens* plants, we saw no buildup of $O_2$ in the rhizosphere detectable on planar optodes, similar to some previous

studies (Waldo et al., 2019). Lai et al. (2012) found significant variation in ROL by species in an assessment of 35 wetland plants, with the highest rates in *Caldesia reniformis* (almost 800 nmol $m^{-2}$ $s^{-1}$), greater than but similar to our highest measurements from *S. americanus*. These high rates out of *S. americanus* suggest a major role in oxidation of the rhizosphere and surrounding anoxic soil, and a significant impact on soil biogeochemistry in regions dominated by this species. However, the lack of ROL seen in mature, darker-colored roots during methylene blue

analysis makes scaling up $O_2$ input to the soil difficult using traditional belowground biomass metrics.

Using previous estimates of photosynthetic rates in *S. americanus* and *S. patens* (DeJong et al., 1982), we can roughly estimate that the rate of $O_2$ released to the rhizosphere represents on average 5-25% of the total maximum fixed $O_2$ for these plants. This high fraction is unlikely to be derived from photosynthesis alone, and thus implies that most of the rhizosphere $O_2$ is derived from gas transport from the atmosphere, as others have found previously

(Colmer, 2003). The increased rate of ROL we see in the daytime likely represents the influence of light and temperature on stomatal conductance, though there may be some contribution of photosynthetically derived $O_2$ as well (Lai et al., 2012). Jiang et al. (2020) found significantly greater ROL among wetland plants in the summer during active plant growth than other seasons, though the magnitude of the difference depended on species; similarly, we see greater ROL in the warmer day period. Extended growing seasons and warmer air temperature

throughout the year will likely increase ROL—but also increase root exudation, as Zhai et al. (2013) found.

### 4.3 Implications for global change

As global change alters plant communities along coasts and elsewhere, there may be unexpected biogeochemical consequences due to species-level plant traits governing ROL and root exudation, including alterations in methanogenesis and methane oxidation, which has further implications for global change. Based on the results of

this study and field data from a long-term experiment (Noyce & Megonigal, 2021), *S. americanus* has a net oxidizing effect on soil, while *S. patens* has a net reducing effect. With sea level rise continuing to result in plant community transition at Kirkpatrick Marsh and other locations in the mesohaline Chesapeake, encroachment of *S. americanus* into *S. patens* areas may reduce methane production due to reduced root exudation, and increased sediment oxygenation. The combination of root exudation rate (higher in *S. patens*) and ROL (higher in *S.*

*americanus*) at the species level likely has significant consequences on methane emissions, as shown in the field (Noyce & Megonigal, 2021; Mueller et al., 2020), which may be exacerbated by increased temperature, as both ROL (Jiang et al., 2020) and root exudation (Zhai et al., 2013) increase with temperature. At the Global Change Research Wetland (GCReW) on Kirkpatrick Marsh, increased temperature has exacerbated the differences in methane emissions between *S. americanus* and *S. patens*-dominated areas (Noyce & Megonigal, 2021), possibly due

to the effect of temperature on rhizosphere processes (Uselman et al., 2000; Zhai et al., 2013). Additionally, elevated $CO_2$ has been shown to increase root exudation in wetland plants (Sánchez-Carrillo et al., 2017), and increases ROL in rice plants (Li et al., 2024), although which process dominates, and has the greatest impact on soil methane



production, has not been studied. The effects of temperature and elevated $CO_2$ combined suggest that differences in

methane emissions occurring as a result of rhizosphere processes may continue to diverge at a species level as global

change progresses, though future research is needed to understand the interplay between these two global change

drivers, along with sea level rise. Understanding plant responses to global change is critical for predicting future

climate feedbacks. Here, we have demonstrated two methods that can be used to assess and quantify root-associated

metabolites and ROL and link plant traits to observed biogeochemical patterns. Further research using these

methods can now be undertaken to quantify how production of root-associated metabolites and ROL will respond to

combinations of elevated $CO_2$, sea level rise, and other consequences of global change.

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

## Competing interests

The contact author has declared that none of the authors has any competing interests.

## Acknowledgements

We are very grateful for the contributions of many people at the Smithsonian Environmental Research Center who helped make this work happen, especially Taylor Smith, Alia Al-Haj, Chelsea Nitsch, Pat Megonigal, Roy Rich,
Selina Cheng, Jamie Pullen, Drew Peresta, Gary Peresta, Carey Pelc, Stephanie Wilson, and Zoe Read. Additionally, we owe thanks to Cynthia Tope-Niedermaier at UMBC who carried out our metabolomics sample processing and annotation. We are also grateful to Jillian and Clark at Delmarva Native Plants for donating plants to this work for use in methods development, and for donating plugs of *S. alterniflora* for use in the root exudation sampling and analyses.

Funding for this project was provided by the U.S. Department of Energy, Office of Science, Office of Biological and Environmental Research, Environmental System Science program under Awards DE-SC0014413, DE-SC0019110, and DE-SC0021112 and the Smithsonian Institution, though the Life on a Sustainable Planet and Postdoctoral Fellowship programs.