# Peer review of "Assessing root-soil interactions in wetland plants: root exudation and radial oxygen loss"

_EGUsphere, 2024_

## Author Response (AR1)

**Dear Reviewers,**

**Thank you both for taking the time to review our paper, and for your helpful and supportive suggestions. Our responses, and actions taken to improve the manuscript, are detailed below in bold. Please note that all line numbers refer to the line number in the track changes version of the revised manuscript.**

Reviewer 1:

In this study, Haviland and Noyce evaluated root exudates and radial oxygen loss (ROL) from wetland plants from a specific marsh. They also conducted an incubation experiment to assess the effect of exudates on $CH_4$ production from soils colonized by specific species. In my opinion, the topic is highly relevant given the current need to understand plant processes that influence greenhouse gas emissions. I consider that the text is clear, provides sufficient detail, and that the experiments were well conducted.

On the weak side of the study is the poor resolution of the ROL profiles provided. The authors clarified the constraints of using planar optodes, noting the low resolution in 2D and the incomplete detection of $O_2$ loss from all roots in a plant. The methylene blue images provided are somewhat insufficient to indicate the sites of ROL. I believe a more detailed evaluation of ROL using other methods (e.g., root sleeving electrodes, Ti-citrate) together with anatomical characterization of roots would provide a more accurate quantification of ROL from roots. Nonetheless, as stated by the authors, for comparison purposes the methods used might still be valid.

**As you highlighted, we present our ROL data largely for comparison between the plants used in our study and other studies with similar methodology. While further quantification of ROL is outside the scope of this project, we agree that the suggested methods would strengthen the community's understanding of this topic. We now address this by expanding our discussion to include the use of more quantitative and precise methods as an avenue for future research (lines 413-415).**

Some aspects that in my opinion will benefit the MS are:

- Including general information on the effect of higher ROL (oxic zones in the rhizosphere) on CH4 oxidation will benefit the text.
- Redox reactions including all the intermediates (i.e., SO4, Fe, Mn..) should be mentioned in the text. The oxidation of such molecules often proceeds at a much higher rate than CH4 oxidation, therefore leaving less O2 for CH4 oxidation. Special consideration should be given to S cycling because of the high sulfate concentrations in the porewater of many intertidal wetland soils.

**Thank you for identifying these gaps and providing helpful suggestions. We have added new language on how oxic zones in the rhizosphere affect CH4 oxidation (lines 362-364) and discuss intermediate redox reactions (lines 58-63).**

Ln 45-48: include more information on how O2 diffusion along and across the roots is affected by factors such as respiration, porosity, root length, tissue density.

**We now include the effects of root anatomy on ROL in the revised manuscript (lines 53-56).**

Adding subtitles to the methods section will make it easier to follow the different methods and techniques applied.

**Thanks for this suggestion, we have added subtitles to the methods.**

Ln 191: is this referred to as for changes in O2?

**You are correct—we meant to say "O2" rather than "R" and have corrected this. Thank you for catching this typo.**

Is Table 2 referring to an average of measured SOD, or is it simply a calculation based on soil porosity and microbial activity? In any case, clarifications are needed on the estimations, and standard deviations should be included next to the averaged values.

**Table 2 represents an estimated rate of ROL from the plants, calculated using the rate of soil oxygen demand (SOD) in a given rhizosphere, and the steady state % of oxygen around a given plant over a time period, as specified in the optodes section of our methodology. We have updated this table to include standard deviations, and have clarified our SOD calculations in the methods (lines 213-215)**

Ln 301-304: The buildup of O2 in darkness appears to contradict with Ln 312-314. In light conditions one should expect higher ROL given photosynthesis (DOI: 10.1007/s11104-015-2382-z). Please clarify.

**Our observed buildup of O2 in darkness was driven by a precipitous decrease in sediment oxygen demand during the night that overshadowed the smaller increase in ROL due to photosynthesis during the day. This drop in SOD is due to the 7 °C difference in temperature between the night and the day period. We have clarified this mechanism in the revised manuscript.**

**Reviewer 2:**

Katherine A. Haviland and Genevieve Noyce investigated root exudation and root oxygen loss from two (four) marsh plant species to test how root oxygen loss and rates of root exudation affect methane production. Root exudates were collected using a combination of soil and hydroponic methods after growing the plants in rhizoboxes, and these exudates were analyzed for total organic carbon (TOC) and metabolites. A 2-D planar optode imaging system was used to measure root oxygen loss. In a separate incubation study, the authors tested the methane production of soil in response to additions of exudate cocktails. The methods and materials used in the study are state-of-the-art and provide valuable insights into the potential consequences of plant community changes in response to sea level rise on methane production, as well as mechanistic insights into how root activity contributes.

Major points:

Exudate Compounds and Methanogenesis: The study does not specify which compounds in the root exudates are responsible for affecting methanogenesis. While the approach of testing natural root exudate cocktails and observing soil response is interesting, it renders the elaborate metabolome analysis somewhat unnecessary, as only speculation on the contributions of specific compounds is possible. Testing single compounds would strengthen the findings.

**We agree completely that testing specific metabolic compounds would strengthen the findings and be a very interesting avenue for future research. Similar work has been carried out on peat previously (e.g. Girkin et al., 2018). One goal of this project was to test methods using the untargeted sequencing approach; now that we are confident it works we will be able to conduct future target approaches, but unfortunately, it is outside the scope of our work at present to do so.**

Global Change Implications: The manuscript could further explore the implications of global change on CO2 production and carbon storage in soil, particularly the relationship between methanogenesis, CO2 production, and soil organic carbon stock changes. For example, could transitions from *Spartina patens* to *Schoenoplectus americanus* lead to reductions in soil organic carbon stocks, even with reduced methane production?

**Thank you for drawing our attention to this, as the ecosystem-level effects of global change (particularly sea level rise) driving a shift from *S. patens* to *S. americanus* are certainly important. Unfortunately, changes in soil organic carbon stock were outside the scope of the data collected for this project, but in response to your comment we have now expanded our discussion section to discuss some potential SOC implications of our results (lines 468-477).**

Methodologies and Clarity: The explanation of the methodologies was not always clear (see minor comments). The manuscript should ensure that all methods are clearly explained and justified. Additionally, figure captions and table headings should be comprehensive and self-explanatory, with all abbreviations defined within the captions to ensure clarity.

**We agree that our methods section could use clarifications and expansion. We now include subheadings in the methods section to help readers follow the methodology of our various experiments and have improved the clarity of our figure captions and table headings.**

Minor line comments refer to clarification, providing methodological details, and the improvement of the scientific relevance:

L9: Opposite relationships and directions need to be named.

**Thank you for pointing that out; we have updated the abstract to clarify which relationships we're referring to (line 9-10)**

L11: The term "reduced" should be clarified to specify if it refers to the entire carbon pool.

**We have revised this language to clarify our meaning (line 12).**

L13: Details on the processes affecting methane emissions and the relevance of rhizosphere processes should be provided. Can rhizosphere processes be measured on the soil surface?

**While we cannot expand substantially in the abstract due to space constraints, this is a valid and interesting question, and we have added language addressing it in the discussion section of our paper (lines 377-381). We have also included more details in the introduction about how rhizosphere inputs affect methane cycling and other biogeochemical processes (lines 46-49).**

L26: The reference to carbon allocation should be updated with newer articles and corrected to accurately cite the given reference, which addresses rhizodeposition including dead organic matter. Assuming half of the photosynthesis carbon being allocated is incorrectly cited from the given reference as rhizodeposition including dead organic matter is addressed in the reference.

**Thank you for bringing this to our attention. We have fix the incorrect reference to half of photosynthesis carbon being allocated and added an additional more recent citation to support our point (lines 27-28).**

L27-29: The sentence regarding O2 sinks, their fuel by root exudates, and redox conditions in soil needs clarification.

**We have revised this sentence to clarify the relationships between O2 sinks, root exudation, and soil redox conditions (lines 28-30) and added additional clarification of O$_2$-driven biogeochemical cycling later in the introduction (lines 60-72).**

L38: The term "sediment biogeochemical regimes" should be defined.

**Thank you for highlighting that this term is unclear; we have changed the term "sediment biogeochemical regimes" to the more specific term "redox conditions" (line 42).**

L60: Soil C storage should be correctly referred to as a stock, not a rate.

**Thank you for catching that; we have fixed this sentence (line 77).**

L73: Clarify how well growth chamber measurements reflect field conditions.

**We agree that this is a limitation of growth chambers, though necessary for certain types of data collection. We have added new text comparing growth chamber experiments to field conditions at Kirkpatrick Marsh in lines 119-121.**

L88: Explain the integration of *Phragmites australis* and *Spartina alterniflora* in the study, given the hypotheses testing only differences between *Schoenoplectus americanus* and *Spartina patens*.

***P. australis* is an invasive species at Kirkpatrick Marsh that is rapidly becoming more dominant and *S. alterniflora* is a common wetland plant species. We included them in this study to provide broader context for the measured ROL of our target species. We have revised lines 90-91 to better clarify this.**

L100: Provide additional details on soil water content and matric potential.

**We have add additional details on water content in the rhizobox soil to the manuscript (lines 112-113).**

L116: Specify in which cases samples are sterile and in which they are not.

**No samples or methods were completely sterile. We have emphasized lines 134-135 to better clarify this.**

L113: Please display root-associated metabolites in Figure 2.

**Thank you for pointing out this incongruence, we have changed the language in Figure 2 to refer to "root" only, rather than root exudates.**

L133: There seems to be a mismatch regarding the filter usage. Was it 0.22 or 0.45 micrometers?

**The filters were 0.22 microns. As you noted, we incorrectly stated in one location that the filters were 0.45 microns and have corrected this.**

L155: Please clarify which plants were used. Is it the same as described in chapter 2.2? If taken from the field, how did you make sure to excavate them entirely?

**The same cohort of plants was used for all analyses. Rather than excavating full plants from the field, we collected rhizomes from the field and grew the plants in a growth chamber. We now clarify this in line 107.**

L160: Please clarify, have the samples been on a shaker for two weeks? Please explain why you chose this procedure.

**Our protocol did include keeping samples on a shaker at a very low RPM for a 2-week period in order to prevent the settling out of sediment from the slurry, to allow for the distribution of the exudate cocktail among the slurry, and to maintain homogeneity of the samples. We have added this explanation to the revised methods in lines 183-184.**

L168: Please add an explanation that justifies the duration of the experiment.

**We have explained the justification of the experimental duration in line 182. The duration of the experiment was based on previous research where synthetic metabolite cocktails were added to peat (e.g. Girkin et al., 2018).**

L184: Were these the same plants as described in Chapter 2.2? If not, please add this information.

**The same cohort of plants was used for all analyses. We now specify this in line 107.**

L339: Methane yield is not presented in Figure 5.

**Thank you for catching this. We have revised the manuscript so that the reference here is for Figure 6, rather than Figure 5 (line 368).**

L339-344: Please clarify the results and how they link to the field scale under changing water levels.

Thank you for drawing our attention to this, as the ecosystem-level effects of global change (particularly sea level rise) driving a shift from *S. patens* to *S. americanus* are certainly important. We have now expanded our discussion section to highlight some potential implications of our results (lines 468-476).

Thank you again to both reviewers for taking the time to review our paper,

Drs. Haviland and Noyce